# Planned Extracorporeal Life Support Employment during Liver Transplantation: The Potential of ECMO and CRRT as Preventive Therapies—Case Reports and Literature Review

**DOI:** 10.3390/jcm12031239

**Published:** 2023-02-03

**Authors:** Cristiana Laici, Amedeo Bianchini, Noemi Miglionico, Niccolò Bambagiotti, Giovanni Vitale, Guido Fallani, Matteo Ravaioli, Antonio Siniscalchi

**Affiliations:** 1Postoperative and Abdominal Organ Transplant Intensive Care Unit, IRCCS Azienda Ospedaliero-Universitaria di Bologna, 40138 Bologna, Italy; 2Department of Medical and Surgical Sciences (DIMEC), University of Bologna, 40126 Bologna, Italy; 3Internal Medicine Unit for the Treatment of Severe Organ Failure, IRCCS Azienda Ospedaliero-Universitaria di Bologna, 40138 Bologna, Italy; 4Hepatobiliary and Transplant Surgery Unit, IRCCS Azienda Ospedaliero-Universitaria di Bologna, 40138 Bologna, Italy

**Keywords:** liver transplant, Veno-Venous Extracorporeal Membrane Oxygenation, Veno-Arterial Extracorporeal Membrane Oxygenation, Continuous Renal Replacement Therapy

## Abstract

Liver Transplantation (LT) has become the gold standard treatment for End-Stage Liver Disease (ESLD). One of the main strategies to manage life-threatening complications, such as cardio-respiratory failure, is Extracorporeal Membrane Oxygenation (ECMO) in the peri-transplantation period, with different configurations of the technique and in combination with other extracorporeal care devices such as Continuous Renal Replacement Therapy (CRRT). This retrospective study includes three clinical cases of planned ECMO support strategies in LT and evaluates their application compared with current literature exploring PubMed/Medline. The three LT supported with ECMO and CRRT were performed at IRCCS Polyclinic S. Orsola-Malpighi, Bologna. All three cases of patients with compromised organ function analysed produced positive outcomes. The planned use of ECMO and CRRT support in peri-transplantation has allowed the patients to overcome contraindications and successfully undergo LT. In recent years, only a few reports have documented successful LT outcomes performed with intraoperative ECMO in critically ESLD patients. However, the management of LT with ECMO and/or CRRT assistance is an emerging challenge, with the need for more published evidence on this topic to guide treatment choices in patients with severe, acute and reversible respiratory and cardiovascular failure after LT.

## 1. Introduction

Liver Transplantation (LT) has become the gold standard for the treatment of End-Stage Liver Disease (ESLD) as well as many other clinical indications. Since the first LT procedure almost sixty years ago, new surgical and clinical techniques have allowed patients with comorbidities or an advanced stage of their disease—which previously would have excluded them from the waiting list—to successfully undergo LT, thus making the procedure accessible to more patients.

In recent years, one of the primary strategies to manage cases with life-threatening complications, particularly cardio-respiratory failure [1,2], has been organ support through Extracorporeal Membrane Oxygenation (ECMO) [3,4,5] in the peri-transplant setting, with different configurations of the technique and combinations with other extracorporeal assistance devices such as Continuous Renal Replacement Therapy (CRRT). While this technique has permitted more patients to receive LT, it is still primarily applied in critical situations when the patient’s life is already at risk rather than as a pre-emptive measure to avoid possible complications before they arise. Yet, planned, preventative cardio-respiratory support in the peri-operative period can sustain vital functions and mitigate the detrimental effect of severe comorbidities, improving the success of LT and facilitating an easier and more effective recovery.

This article aims to present three different clinical cases of planned ECMO-support strategies in LT and assess their application concerning the current literature on the topic.

## 2. Materials and Methods

In this retrospective study, we collected the clinical data of three peculiar cases who underwent LT performed with ECMO support strategies at IRCCS Azienda Ospedaliero-Universitaria di Bologna, Italy, between 1 March 2021 and 31 October 2022. We also reported the number of LT performed in the study period and those requiring ECMO assistance by a multidisciplinary decision approach, including hepatologists, anaesthesiologists, surgeons, perfusionists, cardiologists and nephrologists. The only criterion of inclusion for the use of planned extracorporeal life support was the attending of organ failure reversible after LT. The general indication of the use of ECMO was the presence of risk factors predisposing to intraoperative hemodynamic instability, such as the hemodynamic status of the patient with ESLD and the aetiology or the complications of liver failure, pre-existing cardiovascular disease, the complexity of surgical phases and the risk of reperfusion syndrome with an acute decrease in systemic vascular resistance or new-onset ventricular dysfunction. The choice of a Veno-Venous Extracorporeal Membrane Oxygenation (VV ECMO) was made if there was an elevated risk of respiratory failure in the presence of standard cardiac output. At the same time, we opted for a Veno-Arterial Extracorporeal Membrane Oxygenation (VA ECMO) solution in the presence of respiratory and heart failure after preoperative evaluation for planned ECMO support. The criterion for exclusion of extracorporeal life support was an expected irreversible organ failure despite LT. Instead, in the presence of concomitant Acute Kidney Injury (AKI) we assured intraoperative CRRT support. Being very critical patients, due to the complex clinical status and the risk of cardiorespiratory insufficiency, we only considered the use of non-marginal organs, defined as elderly donors, steatotic livers, donors after circulatory death, and split liver grafts; moreover, we excluded the possibility of living donor LT. The Model for End-Stage Liver Disease (MELD) was routinely used as the score to prioritize organ allocation for LT [6]; however, in our centre, HPS and PoPH represented two standardized MELD policy exceptions according to national and regional organ allocation policies [7]. This study was performed in accordance with ethical guidelines of the World Medical Association’s Declaration of Helsinki and guidelines for Good Clinical Practice. In addition, we conducted a systematic literature review on the PubMed/Medline database, using the following search items: “Liver Transplant”, “Veno-Venous Extracorporeal Membrane Oxygenation”, “Veno-Arterial Extracorporeal Membrane Oxygenation”, “Continuous Renal Replacement Therapy”, “LT”, “ECMO”, “VV ECMO”, “VA ECMO”, and finally, “CRRT”. The decisional process for selecting references is described in Figure 1: the search included all study designs with no language restriction. We also referenced lists of relative articles to ensure that all potential publications on this topic were considered. We excluded studies enrolling children and articles in abstract form without predefined data available. The selection process for references has included 26 articles. Eighteen of the papers described the state of the art about LT, VV ECMO, VA ECMO, and CRRT. Four and three references described VV ECMO and CRRT during LT, respectively, and only one traced the use of VA ECMO in LT. No references were found about the simultaneous use of VV ECMO and CRRT during LT.

Finally, Figure 2 shows the flowchart applied for our three cases.

## 3. Results

The number of LT procedures performed at IRCCS Azienda Ospedaliero-Universitaria di Bologna, Italy, between 1 March 2021 and 31 October 2022, was 185. Of these, we conducted seven with ECMO support strategies, six in VV ECMO and one in AV ECMO. The indications for the use of VV ECMO were five Hepato-Pulmonary Syndrome (HPS), 1 Porto-Pulmonary Hypertension (PoPH), while the only in VA ECMO was a probable group 2 Pulmonary Arterial Hypertension (PAH). Here, we described three peculiar cases of LT performed, respectively, by VV ECMO, VV ECMO combined with CRRT, and finally, VAECMO, their indications and outcomes.

### 3.1. Case 1

A 59-year-old male with alcoholic cirrhosis and a heterozygous S65C mutation in the HFE-related hemochromatosis gene was admitted to our institution to evaluate LT eligibility. The MELD score was 26, and the MELD-Na (MELD-Na score is a variant of the MELD score that includes sodium levels in the calculation) was 30. The patient was also affected by severe primary pulmonary hypertension with associated right ventricular dysfunction; he was dyspnoeic after minimal physical activity (New York Heart Association, NYHA, Class III). Trans-Thoracic Echocardiography (TTE) revealed a dilated and hypokinetic right ventricle, interventricular septum systolic flattening, moderate tricuspidal regurgitation, and right atrial enlargement (right atrial area 32.5 cm^2^) with inferior vena cava dilatation and reduced inspiratory collapse; estimated systolic Pulmonary Artery Pressure (sPAP) was 47 + 10 mmHg. The left ventricle thickness, chamber volumes and contractility were within normal range.

Table 1 summarizes the preoperative right-heart catheterization (RHC) and the metabolic features of three case reports. RHC of case 1 showed precapillary pulmonary hypertension with cardiac index (CI) in the normal range at rest: sPAP was 60 mmHg, diastolic Pulmonary Artery Pressure (dPAP) was 19 mmHg, mean Pulmonary Artery Pressure (mPAP) was 35 mmHg, Pulmonary Artery Wedge Pressure (PAWP) was 6 mmHg, CI was 2.7 L/min/m^2^, Pulmonary Vascular Resistance (PVR) was 4.8 WU, Central Venous Oxygen Saturation (ScvO_2_) was 60%, and finally, hepatic venous pressure gradient (HVPG) was 17 mmHg, confirming clinically significant portal hypertension. According to the European Society of Cardiology (ESC) and the European Respiratory Society (ERS) guidelines, the RHC parameters were suggestive of moderate group 1 PAH [8]. Therefore, based on the European Association for the Study of the Liver (EASL) and the International Liver Transplant Society (ILTS) practice guidelines [9,10], a diagnosis of moderate PAH associated with portal hypertension, commonly referred to as PoPH, was established with the presence of portal hypertension and the lack of alternate causes for PAH such as chronic thromboembolism, chronic lung disease/hypoxia and chronic left heart. At the time of the exam, the patient was received sildenafil 10 mg twice a day and furosemide 80 mg/day. While mild or moderate PoPH should generate MELD exception points and facilitate accession to LT, severe or unresponsive PoPH is a relative contraindication for LT. Thus, anti-PAH therapy was supplemented by increasing the dose of sildenafil to three daily doses and adding another agent, selexipag 400 mcg, twice daily. After three weeks, this therapy decreased mPAP from a threshold value of 35 mmHg, corresponding to moderate PoPH, to 32 mmHg, reaching mild PoPH. Additionally, other RHC parameters improved: PVR value was 2.9 WU, CI was 3.5 L/min/m^2^ and ScvO_2_ was 65%. In addition, preoperative chest X-rays showed a bilateral pleural effusion.

LT required VV ECMO to provide respiratory function support to optimize intraoperative and postoperative Delivery of Oxygen (DO_2_). We opted for VV ECMO than VA ECMO for the elevated risk of respiratory failure during LT in the presence of standard cardiac output. Basal values of blood gas analysis showed partial pressure of oxygen in the following: arterial blood (PaO_2_) 70 mmHg, P/F ratio of PaO_2_ to fractional inspired oxygen (FiO_2_) 141, Haemoglobin (Hb) 8.1 g/dL, Lactate 1.5 mmol/L, pH (potential of Hydrogen) 7.4, partial pressure of carbon dioxide (PCO_2_) 38 mmHg, Bicarbonate (HCO_3_-) 24 mEq/L, Base Excess (BE) −0.3. After the induction of general anaesthesia, we started VV ECMO through a 21 French (Fr) cannula of drainage in the right femoral vein and a 19 Fr cannula returning to the right internal jugular. Table 2 shows the hemodynamic monitoring used during LT in VV and VA ECMO.

During surgery, we adopted a Swan Ganz catheter for invasive hemodynamic monitoring. We used a continuous infusion of milrinone and inhaled Nitric Oxide (NO) (maximum dose 14 ppm) to reduce PVR and norepinephrine (maximum dose 0.6 mcg/kg/min) to maintain Main Arterial Pressure (MAP) > 65 mmHg. Anticoagulation was not necessary because Activated Clotting Time (ACT) values were consistently above 200 s. The intraoperative course was uneventful.

After LT, the patient recovered in the Intensive Care Unit (ICU) with an ECMO flow of 3.5 L/min. Milrinone infusion and NO were suspended, and the previous pulmonary hypertension therapy resumed. The patient was weaned from mechanical ventilation and extubated 10 h after surgery; oxygen therapy with High Flow Nasal Cannula (HFNC) was then adopted. ECMO was suspended 4 h later.

On Postoperative day (POD) 2, the patient developed an AKI, Stage 2 Acute Kidney Injury Network, which responded quickly to the reduction in diuretic infusion and to balanced crystalloid solution infusion.

High-Resolution Computed Tomography performed in POD 5 demonstrated an increase in both right and left pleural effusion with a complete atelectasis of both lower lobes. A right thoracic drainage was then placed and a left evacuative thoracentesis performed, resulting in a rapid improvement in the patient’s respiratory function. Due to the patient cardiologic status, to avoid fluid overload and consequent systemic venous congestion, a furosemide continuous infusion was started postoperatively, and a negative fluid balance was maintained. Norepinephrine was de-escalated until suspension in POD 5. In POD 8, the patient was discharged from our ICU to the surgical ward, and in POD 35, he left the hospital. However, at the last follow-up, three months after LT, a new RHC, compared with the previous one, had documented persistent pre-capillary pulmonary hypertension with a reduced CI at rest and right heart failure (CVP 17 mmHg, mPAP 42 mmHg, PAWP 12 mmHg, CI 1.9 L/min/m^2^, PVR 7.5 WU).

### 3.2. Case 2

A 45-year-old female patient was admitted to our department to evaluate her eligibility for LT. She suffered from liver failure due to alcoholic cirrhosis. Her clinical history included severe haemolytic spur cell anaemia, which required multiple blood transfusions. At the time of her first diagnosis, her MELD score was 22. The liver function then deteriorated markedly during a new episode of haemolysis (Hb 6.4 g/dL), causing AKI and consequently increasing the MELD score to 31 and the MELD-Na score to 33 (Bilirubin 7.9 mg/dL; International Normalized Ratio, INR, 2.06; Creatinine 2.46 mg/dL). The progression of AKI to anuria led to congestive heart failure with pulmonary hypertension. A cardiac ultrasound revealed the left ventricle to be of standard size and wall thickness, the absence of segmental kinetic deficits, and a pattern of restrictive left ventricular filling (the ratio of peak velocity blood flow from left ventricular relaxation in early diastole to peak velocity flow in late diastole caused by atrial contraction, E/A wave, 120/60 cm/sec: deceleration time 118 ms). On the right ventricle, the ultrasound revealed dilation with moderate to severe tricuspid insufficiency, and an estimated sPAP of 45 mmHg. The patient was invasively monitored with pulmonary artery catheters, resulting in high central pressure values of the venous and pulmonary wedge (up to 30 and 22 mmHg, respectively). We started a Continuous Veno-Venous Hemodiafiltration (CVVHD) with regional citrate anticoagulation by removing fluids at the speed of 120–150 cc/h, finally obtaining a persistent negative fluid balance with regression of organ congestion. Secondary hypotension required the infusion of vasopressors with norepinephrine up to 0.3 mcg/kg/min. After three days of renal replacement therapy (RRT), due to the onset of signs of citrate accumulation (a total calcium-to-calcium ion ratio equal to 2.8), we changed the dialytic modality, switching to CVVHD without anticoagulant.

Severe volume overload was managed by subtractive haemofiltration therapy, which reduced the central venous pressure (CVP) and the PAWP (down to 12 and 15 mmHg, respectively). Still, the patient remained profoundly hypoxemic (PaO_2_ 45–55 mmHg) with severe intrapulmonary shunting. We then performed extensive testing, including high-resolution and computed tomography pulmonary angiography. Still, we did not identify the alternative cause of hypoxemia apart from trivial pleural effusions and some dependent lung collapse. Given the persistent hypoxemia and an A-a gradient of 18 mmHg in a patient with acute on chronic liver failure, a bubble echocardiogram, was performed to confirm/rule out a HPS. The exam revealed the presence of the bubbles in the left ventricle within 6 cardiac cycles, thus confirming HPS. Despite all efforts to improve the shunt flow (to prevent hypovolemia), including adjustments to the ventilator settings, the patient remained hypoxemic. The only curative treatment for HPS is LT, resulting in a significant improvement in HPS in more than 85% of cases affected by severe hypoxaemia, thanks also to the introduction of HPS as a MELD exception [7].

Therefore, not being able to correct for the hypoxemia in any other way, after five days of CRRT, we performed LT using the piggyback technique with VV ECMO, without interrupting the administration of CRRT that the patient had been continuously undergoing since first being admitted to the ICU. The VV ECMO was performed under fluoroscopic or Trans-Esophageal Echocardiography (TEE). A 21 Fr cannula for venous drainage in the right femoral vein was inserted, as well as a 19 Fr cannula returning to the right internal jugular, maintained by a perfusionist who regularly monitored the flow of the circuit. The position of the VV ECMO and CVVHD cannulas used during LT is illustrated in Figure 3.

At the end of the surgery, the graft function remained well preserved, and the pulmonary shunt flow decreased according to a semiquantitative echocardiographic and clinical assessment. As a result, the VV ECMO circuit was removed after 36 h. After 24 h, the patient was extubated. In the absence of instability, on the fourth postoperative day, CVVHD was stopped, and the patient was discharged from the ICU without supplemental oxygen therapy. The patient’s last follow-up was 12 months after LT, with kidney and liver function tests in normal ranges. Concerning the last cardiological follow-up, the patient had a NYHA class I and persistently resolved hypoxia.

### 3.3. Case 3

A 52-year-old female patient was admitted to our department to evaluate LT eligibility. She presented liver failure due to Hepatitis D and B virus-related chronic cirrhosis with a MELD score of 22. Her history included moderate rheumatic mitral stenosis associated with severe mitral regurgitation, biatrial dilatation with pulmonary hypertension and severe tricuspid regurgitation. Moreover, she had refractory atrial fibrillation with a rapid ventricular response and chronic hypotension (MAP < 60 mmHg). Liver and kidney function progressively deteriorated after an episode of bacterial peritonitis, increasing the MELD score to 35. Surgical mitral valve correction was contraindicated due to her liver failure and LT was contraindicated due to severe cardiopathy.

Upon ICU admission, a cardiac ultrasound revealed an ejection fraction (EF) of 40% with no segmental kinetic deficits, dilation of the right ventricle with moderate-to-severe tricuspid insufficiency and an estimated sPAP of 65 mmHg. The PVR value was 4 WU. Since PAH can often be a complication of certain left heart failures, such as valvular heart diseases and congenital defects, we classified case 3 as a probable group 2 PAH worsening from the cirrhotic status of the patient [6]. A fluid challenge test demonstrated poor cardiac tolerance with an increase in PAWP up to 20 mmHg, and CVP up to 22 mmHg. Considering the low tolerance to fluids and the high risk of fluid overload combined with low EF, we decided to use VA ECMO support during LT.

Cannula placement was performed with TEE and fluoroscopy guidance. The two draining cannulas were positioned in the right internal jugular vein (19 Fr) and the right femoral vein (21 Fr), reaching below the renal veins; a 15 Fr inflow arterial cannula was positioned in the right femoral artery. An additional 8 Fr arterial catheter was placed distally in the femoral artery to prevent right leg ischemia. Limb perfusion was carefully evaluated using a pulse oximeter and Doppler ultrasound. Coagulative parameters were assessed by ACT and thromboelastography (TEG) every 30 min and corrected with heparin.

VA ECMO flow was kept between 0.9 and 2.1 L/min/m^2^. Norepinephrine (maximum dose 0.2 mcg/kg/min) was administered after the induction of anaesthesia, maintaining MAP > 65 mmHg. Intraoperative hemodynamic monitoring was achieved by Swan Ganz catheter, TEE and pulse contour analysis.

Cardiac function remained adequate during surgery without increased pulmonary pressures or pulmonary oedema. During liver reperfusion, we witnessed no right-heart failure nor hepatic congestion and, after reperfusion, the patient showed progressive lactate clearance. At the end of the surgery, the improvement in pulse pressure, the lack of inotropic drugs and the increase in EF to 52% (measured with TEE) facilitated rapid weaning from VA ECMO 6 h after surgery.

After 24 h, norepinephrine was stopped, and the patient was extubated. Although graft function gradually improved, due to AKI we started a CVVHD with regional citrate anticoagulation. We removed fluids at the speed of 120–150 cc/h, finally obtaining a persistent negative fluid balance to avoid organ congestion. CVVHD was continued for ten days to keep strict control of fluid balance, solute removal and better hemodynamic stability.

A high-flow arteriovenous fistula was found 15 days after ICU discharge between the right femoral artery and the right femoral vein and was surgically treated under spinal anaesthesia. The echocardiographic parameters showed a progressive improvement in cardiac function in the days following surgery. After ten days, EF was 50% with no dilation of the right ventricle and moderate tricuspid insufficiency. On day 15 post-LT, EF was 55%, and the patient was discharged from the ICU. Ten months after LT, the patient successfully underwent mitral valve replacement with simultaneous tricuspid annuloplasty. Two years after the LT, our 52-year-old female patient is alive with good liver and cardiac function.

Finally, Table 3 summarizes the clinical features and outcomes of three patients who underwent LT.

### 3.4. Literature Review

#### 3.4.1. Veno-Venous Extracorporeal Membrane Oxygenation in OLT

VV ECMO primarily supports lung function by taking up blood from the venous compartment and returning oxygenated venous blood into the same venous sector, acting as an artificial lung in tandem with the normal lungs. The oxygenated blood mixes with the native venous return such that the resultant PaO_2_ and saturation represents a mixture of the oxygenated extracorporeal blood and the unoxygenated venous blood. This technique makes it possible to control the arterial oxygen content and, in the presence of standard cardiac output (CO), provide optimal DO2 to support tissue metabolism [11].

The main indication for VV ECMO is rescue therapy for reversible severe respiratory failure refractory to optimal mechanical ventilation and medical treatment [11,12]. In LT HPS with severe hypoxemia refractory to standard therapy could be a possible indication, as it may take weeks to improve after transplantation, and VV ECMO has been used in this setting as a bridge therapy [10]. Some reports also concern ECMO use after LT as a rescue strategy in patients with severe cardiopulmonary failure [2,13].

Monsel el Al. [14] described a case where VV ECMO was successfully used for five days as a bridge therapy to permit uncomplicated LT surgery on VV ECMO in an alcoholic cirrhotic patient with acute respiratory distress syndrome aggravated by HPS, which otherwise would not have been possible. Frank et al. described a case of reLT performed with a rescue ECMO assistance in a patient with hepatitis B cirrhosis that experienced initial graft thrombosis and pulmonary embolism necessitating placement on VV ECMO, maintained while receiving re-transplantation successfully [15].

With a veno-venous configuration, the patient relies on his hemodynamics, such that CO, PVR and systemic vascular resistances (SVR) are unchanged during extracorporeal gas exchange [11]. This situation may only sometimes be true; for instance, beneficial effects on right-ventricular function due to significant decreases in mPAP after initiating VV ECMO have been reported. These effects could be due to a reduction in hypoxic pulmonary vasoconstriction and PaCO2 [16,17].

However, to our knowledge, cases of planned ECMO in patients with HPS during LT have not been reported.

#### 3.4.2. Veno-Venous Extracorporeal Membrane Oxygenation and Continuous Renal Replacement Therapy in OLT

AKI is a common finding in patients with ESLD with an estimated prevalence of 20% at the time of transplantation; 10% to 30% of these patients require RRT [18].

LT is often associated with haemodynamic instability, altered acid–base and electrolyte balance and extensive and wide fluids shifts. In many cases, despite close haemodynamic and metabolic monitoring, the intraoperative course is complicated by sudden haemodynamic instabilities, resulting in the deterioration of kidney function. For this reason, massive amounts of blood and fluids are required to correct coagulopathies and dysionias. Lactic acidosis can result from the clamping of large vessels and hyperkalaemia is common after reperfusion of the graft and numerous transfusions of blood elements. These intraoperative complications are exacerbated in patients with renal failure and traditional strategies may be ineffective.

The rationale behind the use of intraoperative RRT is to overcome complications due to haemodynamic, coagulative, and metabolic decompensation during LT. There are numerous sources in the literature describing cases of the use of RRT during LT.

Nadim et al. [18] conducted a retrospective study of patients undergoing intraoperative haemodialysis from 2002 to 2012. Seven hundred thirty-seven patients underwent transplantation, of which 32% received intra-operative haemodialysis. This is the first large study to demonstrate the safety and feasibility of intraoperative haemodialysis in a cohort of critically ill patients with high MELD scores undergoing LT, with good patient and kidney outcomes.

Townsend et al. [19] performed a retrospective review of adult patients who received intraoperative CRRT during LT at the University of Alberta Hospital, between 1 January 1996 and 31 December 2005. The survival rate was 97.6% at 1 month and 75.6% at 1 year. Recovery of renal function to RRT independence occurred in 100% of survivors within 1 year. According to the data, intraoperative CRRT during LT seems to be feasible, safe and may act as an adjuvant therapy for patients with preoperative AKI.

Negai et al. [20] emphasize the use of continuous VV hemofiltration (CVVH) with dilutional methods of sodium correction to manage severe hyponatremia (<125 mEq/L), which allows electrolyte disturbances to be corrected gradually. In all the three cases described, intraoperative and postoperative serum sodium levels were successfully managed, thus preventing neurological complications and other significant postoperative morbidities.

#### 3.4.3. Veno-Arterial Extracorporeal Membrane Oxygenation in OLT

VA ECMO is primarily used for reversible cardiovascular collapse post-LT reperfusion secondary to massive pulmonary embolism, intra-cardiac thrombus, right heart failure, or air embolus. VA ECMO can also decrease hepatic congestion during and after LT [21,22,23]. The use of ECLS (Extra Corporeal Life Support) in adult LT is increasing, also as a preoperative or intraoperative intervention to overcome possible contraindications of LT.

Lauterio et al. described an emergency intraoperative implantation of ECMO for refractory cardiogenic shock secondary to myocardial ischemia during LT as a bridge to myocardial surgical revascularization by two coronary bypasses [24].

A few cases have been found in the literature in which VA ICMO is used as a planned strategy in stable patients, such as the case reported by Barbas et al. [25], who describe a case about the use of VA ECMO in a LT patient with pulmonary hypertension refractory to pharmacological treatment, which would have contraindicated the transplantation.

Sun et al. described a case of a patient affected by HBV-liver cirrhosis (MELD 24) with severe mitral regurgitation, severe tricuspid regurgitation, left atrium and left ventricle enlargement, cardiac insufficiency, pulmonary arterial hypertension, and hypoxemia. VA-ECMO was planned during LT, seeing that the patient might not tolerate a large amount of fluid load during the perioperative period and to prevent reperfusion syndrome. Thirty hours after LT, the ECMO was removed, and the respiratory support was eighteen hours later [5]. Martucci et al. reported a case of rescue VA ECMO for postreperfusion cardiac arrest due to right ventricular dysfunction caused by PoPH recognized during LT. VA-ECMO restored general perfusion and protected the graft from venous congestion, but a septic shock occurred [3].

These patients were at high risk of developing heart failure, so the choice fell on a support such as VA ECMO. In pulmonary hypertension, the use of VA ECMO as peri-procedural support is analogous to the application of VV ECMO during LT. Furthermore, if a cardiocirculatory failure occurs, VA ECMO provides immediate and full hemodynamic support, preventing irreversible instability. VA ECMO is in opposition to VV ECMO, which is only valid when cardiac function is preserved.

As reported in case 3, using VA ECMO allowed us to overcome the patient’s cardiopathy during the transplantations. No similar cases have been found in the literature. VA ECMO has been used to support the patients’ cardiac function, which was compromised due to her severe valvopathy, and to successfully transplant her with an uncomplicated postoperative course.

## 4. Discussion

Pre-existing conditions and surgical complications can compromise a patient’s eligibility for LT and exclude the patient from the waiting list. ESLD can cause several underlying and overlapping pathophysiological conditions that contribute to intraoperative hemodynamic instability, such as vasodilation, hyperdynamic circulatory profile, and cirrhotic cardiomyopathy. Likewise, the surgery can cause life-threatening hemodynamic instability, namely, an acute reduction in preload (instant decrease of more than 80% of venous return) due to total caval clamping, massive bleeding, and acute drainage of ascites. These occur during the anhepatic phase and are treated with a rapid fluidic filling. The fluid challenge in this step will further increase the venous return of the next stage. The surgical phase also poses the risk of acute increase in preload (instantaneous) after graft reperfusion and portal vein reperfusion after prolonged vein clamping in patients with portal hypertension (in addition to inferior vena cava reperfusion, usually providing 80% of venous return). Lastly, an overflow of the portal vein with a reduction in the splanchnic venous tension can compromise the patient’s ability to undergo LT successfully. Portal overflow alone can cause pulmonary oedema and acute pulmonary hypertension, such as after the placement of a transjugular intrahepatic portosystemic shunt, even in the absence of evident heart disease [26]. Another life-threatening hemodynamic change is the post-reperfusion syndrome, defined as severe hypotension requiring vasopressor infusion associated with hemodynamically significant arrhythmias (incidence 4 and 81%, respectively), which is also correlated to an acute decrease in SVR or ventricular dysfunction due to ischemia–reperfusion injury [27]. Other intraoperative complications include the acute collapse of SVR after graft reperfusion, rapid alterations in electrolyte and acid/base balance, rapid hypothermia and malignant arrhythmias triggered using catecholamines [28]. All these factors put the patient at too high a risk of cardio-respiratory organ failure to undergo LT without the planned application of organ support successfully.

Both the first and second cases had severe refractory hypoxemia due to PoPH and HPS, respectively. In both situations VV ECMO can help mitigate the risk of hypoxemia by improving proper oxygenation, especially during LT, with the second case having acute renal failure treated with RRT two days before LT. However, given the severity of the ESLD, both patients’ short-term chance of survival without LT was extremely low, but the risk of life-threatening complications during LT was extremely high. Therefore, we decided to use peri-operative VV ECMO to manage possible hypoxemia better, thereby preventing pulmonary vasoconstriction and ventilation-perfusion mismatch.

Moreover, the acute complications of the second patient (water overload, hyperkalaemia, acidosis) required us to maintain CRRT alongside VV ECMO. The use of CVVH offered significant important advantages: (1) it reduced the risk of water overload and the consequent congestion of the lungs and of other organs (including the implanted organ itself); (2) it reduced the risk of hyperkalaemia—linked to anuria and aggravated by cytolysis—which can cause surgical tissue damage, the release of intracellular solutes into circulation and, consequently, serious arrhythmias or cardiac arrest; (3) it facilitated the better management of lactic acidosis, especially during the anhepatic phase.

While these organ-support devices can mitigate these complications, they also pose new management questions. One of the main issues arising from using VV ECMO-CRRT in patients affected by liver failure is the increased risk of bleeding during transplantation. Therefore, anticoagulation therapy with standard heparin is necessary for VV ECMO to operate smoothly and monitor the ACT levels; values around 180 s are considered acceptable to avoid bleeding [29]. Although platelet count can be accurately measured, this does not indicate the quality, which is significant because progressive platelet dysfunction occurs with increasing ECMO time; isoelastic tests (TEG or rotational thromboelastometry, ROTEM) have the potential to provide accurate and timely identification of coagulopathy, identifying both hypo- and hypercoagulopathy [30].

Furthermore, the potential disadvantages of CRRT with a circuit separate from VV ECMO are worth assessing when designating a life-support plan. Most notably, (1) the management of two circuits requires a perfusionist and a nurse specialized in dialysis, and (2) the greater length of the tubes can disperse heat and consequently increase coagulation in the circuit and cause hypothermia.

However, there are advantages to be noticed, namely, the ability to independently manage volume control, which is a critical issue for patients assisted with ECMO undergoing LT. Since a volume change can occur rapidly during the transfusion operating procedures, CRRT might be helpful in this setting, along with a dialysis nurse present whenever possible. In addition, two separate circuits provide insurance in case of malfunction; it allows the CVVH circuit to be immediately substituted without interrupting patient oxygenation. Moreover, for vulnerable patients with a short window of time for LT who cannot wait for other high-risk contraindications to be managed, the planned, intraoperative application of CRRT and ECMO can allow them to undergo LT even before fully resolving otherwise compromising health factors.

Regarding the third clinical case, the patient was considered ineligible for LT due to severe heart failure, responsible for the group 2 PAH. The subject presented moderate mitral stenosis associated with severe mitral regurgitation, biatrial dilatation, pulmonary hypertension, and severe tricuspid regurgitation with EF < 40%. Given these conditions, the probability of intraoperative cardio-circulatory complications was unacceptably high. We also predicted that the patient would be incapable of tolerating rapid fluctuations in preload, afterload, and heart rate. She would also likely suffer reperfusion syndrome during LT and post-operative graft dysfunction.

In this case, the purpose of VA ECMO was to provide temporary cardiopulmonary support in anticipation of a refractory shock during the various surgical phases and to preserve graft function. Furthermore, the machine could have provided life support in case of cardiac arrest, which was highly probable in this scenario.

Key issues must be carefully considered when applying ECMO for LT:(1)Although axillary artery cannulation has advantages such as a lower risk of limb ischemia and atheroembolisation, it also carries a higher rate of bleeding and infection. In addiction it requires a surgical approach to establish the vascular access [31]. We chose arterial femoral percutaneous ultrasound-guided cannulation using the Seldinger technique to reduce the bleeding risk and limit the ischemic organ time. We determined the best diameter of the cannulas to achieve adequate circulatory support using an ultrasound evaluation of the vessels.(2)Since oxygenated and decarboxylated blood is returned to the arterial system in a retrograde way, it creates a unique hemodynamic phenomenon. The retrograde flow associated with cardiac dysfunction can cause blood stasis and thrombus formation in the left ventricle [13,32]. To avoid these problems, we monitored the aortic valve movement and excluded intracardiac thrombi using TEE. We also limited amines and used two venous drainage cannulas to ensure the best possible drainage with a reduction in left ventricle preload. We maintained diuresis around 1 mL/kg/h with furosemide 10 mg/h during the LT to reduce preload.(3)To help the surgeon feel the pulsatility of the hepatic artery during the reconstruction of the vessel, we reduced the ECMO flow. The maintenance of peripheral arterial pulsatility during ECMO is also essential to prevent renal failure, a frequent complication among these patients. However, it is essential to underline that the mechanisms of AKI in patients treated with ECMO are complex and multifactorial [33].

In our patient, we think that the main factors causing AKI include the worsening of the already-impaired renal function (hepatorenal syndrome), the hypotension before initiation of the ECMO support, the presence of renal congestion during surgery and the ischemia–reperfusion injury of the graft.

Without VA ECMO support, this patient was trapped in a Catch-22. She could not undergo LT because of her cardio-vascular contraindications, which required valve correction surgery. However, she could not experience valve correction surgery because of her liver disease. However, thanks to VA ECMO support, the patient was able to successfully receive a LT and was then able to undergo valve correction surgery ten months after the transplant.

Identifying ECMO procedures and drivers’ cost is essential information to clinicians and hospital management. Therefore, it could be an interesting analysis to conduct in the future, also considering the risk/benefit ratio of the procedure itself.

Although we have yet to conduct a cost-effectiveness analysis of ECMO use during LT, the economic investment in the use of planned extracorporeal life support was reduced by its sharing between departments of our Institution specialized in solid organ transplantation and cardiac surgery. Furthermore, it is amply repaid by the saved days of hospitalization for patients with ESLD who cannot otherwise be transplanted and by the costs associated with the life-saving management of severe liver failure complications.

In conclusion, ECMO can play an essential role in ensuring the security of the liver recipients during the surgery and in the postoperative period. However, its standardized use is challenging in all transplant centres due to the question of resources, experience, and complexity of the cases. In the absence of reliable data in the literature, this approach can be recommended in centres with large volumes of transplants, where a multidisciplinary team with transversal skills can decide on a case-by-case basis. Furthermore, ECMO should be considered in particular clinical situations where the risk of hepatic congestion and graft ischemia–reperfusion injury is high, such as in our described cases of PoPH or HPS, and potentially reversible after LT.

## 5. Conclusions

Evidence for the effective use of emergency extracorporeal life support during LT has already been well established. This multidisciplinary strategy makes it possible to progressively expand the pool of ESLD patients eligible for LT and related indications. However, further studies and clinical assessments are needed to understand better the risks and benefits of preventative extracorporeal life support so that the transplant team can assess the planned use in operations.

All three cases analysed in “IRCCS Azienda Ospedaliero-Universitaria di Bologna”, produced positive outcomes, allowing patients with compromised organ function to undergo transplantation successfully. We believe further studies could help determine a criterion of applicability of ECLS for planned uses, in particular for VA ECMO, which is supported by few cases in the literature.

## Figures and Tables

**Figure 1 jcm-12-01239-f001:**
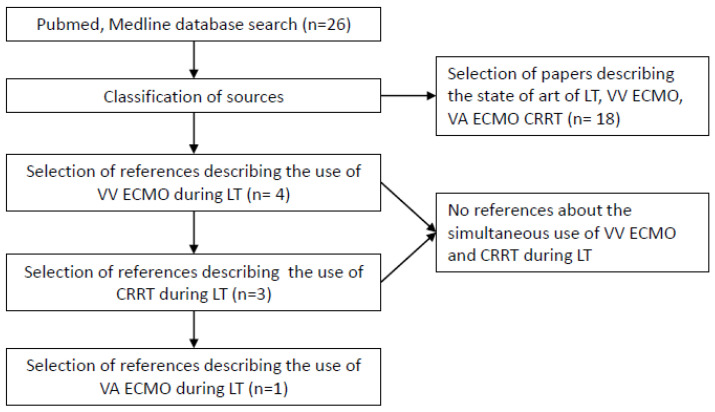
Selection process for references included. VV ECMO: Veno-Venous Extracorporeal Membrane Oxygenator; VA ECMO: Veno-Arterial Extracorporeal Membrane Oxygenator; CRRT: Continuous Renal Replacement Therapy; LT: Liver Transplantation.

**Figure 2 jcm-12-01239-f002:**
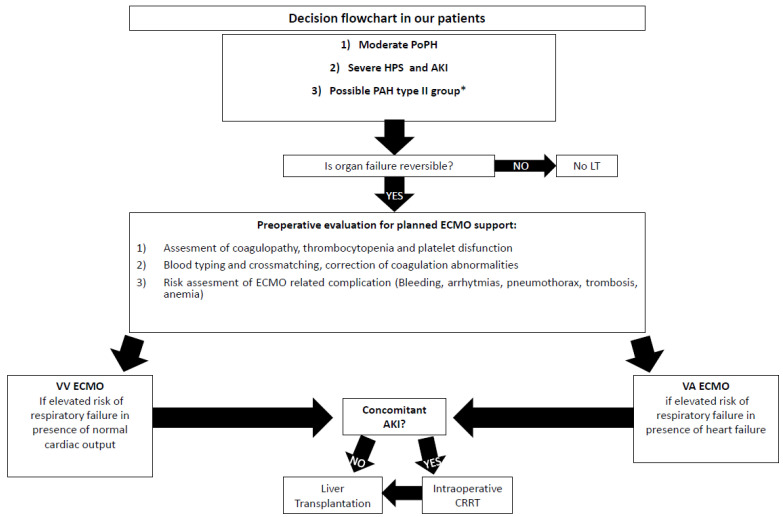
Decision Flowchart in our cases. * WHO Group 2 encloses PAH due to left heart disease. Since PAH can often be a complication of certain left heart failures, such as valvular heart diseases and congenital defects, we classified case 3 as a probable group 2 PAH worsening from the cirrhotic status of the patient [8]. HPS: Hepato-Pulmonary Syndrome; PoPH: Porto-Pulmonary Hypertension; PAH: Pulmonary Arterial Hypertension; ECMO: Extracorporeal Membrane Oxygenator; VV ECMO: Veno-Venous Extracorporeal Membrane Oxygenator; VA ECMO: Veno-Arterial Extracorporeal Membrane Oxygenator; AKI: Acute Kidney Injury; EF: Ejection Fraction; LT: Liver Transplantation; WHO: Word Health Organization.

**Figure 3 jcm-12-01239-f003:**
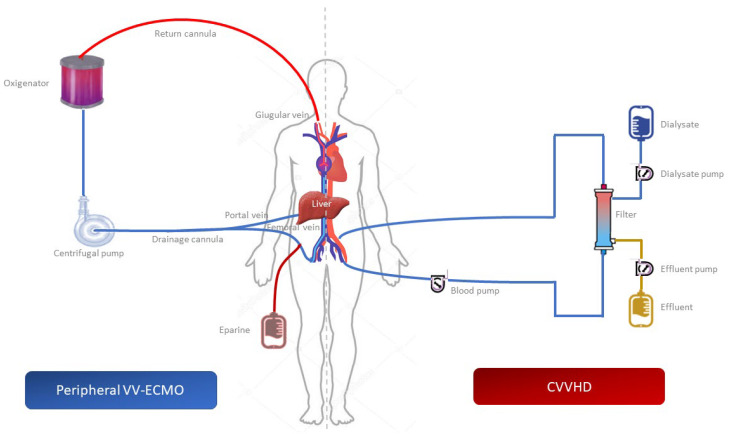
Graphic scheme of Veno-Venous Extracorporeal Membrane Oxygenation (VV ECMO) and Continuous Veno-Venous Hemodiafiltration (CVVHD).

**Table 1 jcm-12-01239-t001:** Summary of hemodynamic and metabolic parameters.

Hemodynamic Parameters	Case 1 PoPH	Case 2 HPS and AKI	Case 3 Possible PAH Type II Group
PAM mmHg	>60	>60	<60
CI L/min/m^2^	3.5 *	5.8	3.2
PAWP mmHg	6	15 **	20 ***
CVP mmHg	8	12 **	22 ***
sPAP mmHg	60 *	40 **	65
dPAP mmHg	19	22 **	24
mPAP mmHg	32 *	28 **	37
PVR WU	2.9 *	1.8	4.0
EF %	>40	>40	40
**Metabolic Parameters**	**Case 1 PoPH**	**Case 2** **HPS and AKI**	**Case 3** **Possible PAH type II group**
pH	7.4	7.38	7.32
PaCO_2_ (mmHg)	38	35	36
PaO_2_ (mmHg)	70	45–55 with severe intrapulmonary shunting	60
HCO_3_- (mmol/L)	24	25	22
BE (mmol/L)	−0.3	−0.4	−1.2
Lactate (mmol/L)	1.5	0.8	1.4
P/F ratio	<150	<200	<200

* After increased sildenafil and added selexipag 400 mcg twice daily; ** After CVVHD by removing fluids; *** After a fluid challenge test. PoPH, Porto-Pulmonary Hypertension; HPS, Hepato-Pulmonary Syndrome; AKI, Acute Kidney Injury; PAH, Pulmonary Arterial Hypertension; PAM, Mean Arterial Pressure; CI, Cardiac Index; Pulmonary Artery Wedge Pressure; CVP, Central Venous Pressure; sPAP, systolic Pulmonary Artery Pressure; dPAP, Diastolic Pulmonary Arterial Pressure; mPAP, mean Pulmonary Artery Pressure; PVR, Pulmonary Vascular Resistance; EF, Ejection Fraction; mmHg, millimetre of mercury; L, litres; m, minutes; m^2^, square meter; pH, potential of Hydrogen; PaCO_2_, partial pressure of carbon dioxide in arterial blood; PaO_2_, partial pressure of oxygen in arterial blood; HCO_3_-: Bicarbonate; BE, Base Excess; P/F ratio, PaO_2_/FiO_2_; FiO_2_, partial pressure of oxygen fraction of inspired oxygen; CVVHD, Continuous Veno-Venous Hemodiafiltration.

**Table 2 jcm-12-01239-t002:** Intraoperative monitoring during VV and VA ECMO.

Intraoperative Monitoring Devices during ECMO	Output	Type of ECMO
Pulmonary artery catheter	Monitoring of SvO_2_, cardiac index, central venous pressure, wedge pressure, systemic vascular resistance, pulmonary vascular resistance	VV and VA
Arterial catheter	In left and right radial arteries, to monitor invasive blood pressure and bilateral PaO_2_	VV and VA
Pulse Oximetry	In both feet to detect right leg hypoperfusion and on the right hand to detect Harlequin syndrome	VV and VA
Neurologic monitoring	Compare left vs. right cerebral perfusion (NIRS) and cerebral activity (BIS)	VA
Trans-Esophageal Echocardiography	ECMO cannulas placement, Titrate ECMO support, ECMO weaning, Hemodynamic monitoring, fluid balance	VV and VA

VA: Veno Arterial; VV: Veno Venous; ECMO: Extracorporeal Membrane Oxygenator; SvO_2_: Mixed Venous Oxygen Saturation; PaO_2_: Partial pressure of Oxygen in arterial blood; NIRS: Near-Infrared Spectroscopy; BIS: Bispectral Index.

**Table 3 jcm-12-01239-t003:** Summary of clinical case results.

Patient	Age/Gender	LT Indication	ECMO Indication	Mode/Type of Cannula	Timing/Duration of Cannulation	Outcome
1	59/Male	Alcoholic Cirrhosis	Moderate PoPH	VV ECMO/21 Fr cannula in the right femoral vein and a 19 Fr cannula in the right internal jugular vein.	After induction of the anaesthesia/14 h	Alive after three months from LT
2	45/Female	Alcoholic Cirrhosis	Severe HPS and AKI	VV ECMO and CVVHD/21 Fr cannula in the right femoral vein and a 19 Fr cannula in the right internal jugular. For CVVH cannula 12 FR in the left internal jugular vein.	Preoperative 3 days CVVHD, VV ECMO after induction of the anaesthesia/stop VV ECMO after 36 h the end of the surgery, stop CVVHD in POD 4 postoperative day	Alive after two years from LT; the last follow-up with the patient took place 12 months after LT, with kidney and liver function tests in normal ranges.
3	52/Female	HDV/HBV-related Cirrhosis	Probable group 2 PAH	VA ECMO/19 Fr cannulas in the right internal jugular vein, 21 Fr in the right femoral vein and 15 Fr cannula in right femoral artery. An additional 8 Fr arterial catheter in the femoral artery	After induction of the anaesthesia/6 h	Alive two years after LT. Ten months after liver transplant, replacement mitral valve with simultaneous tricuspid annuloplasty.

LT: Liver Transplantation; ECMO: Extracorporeal Membrane Oxygenation; HPS: Hepato-Pulmonary Syndrome; PoPH: Porto-Pulmonary Hypertension; PAH, Pulmonary Arterial Hypertension; Fr: French; VV ECMO: Veno-Venous Extracorporeal Membrane Oxygenation; VA ECMO: Veno-Arterial Extracorporeal Membrane Oxygenation; CVVHD: Continuous Veno-Venous Hemodiafiltration.

## Data Availability

Data supporting reported results can be found at IRCCS Azienda Ospedaliero-Universitaria di Bologna, department of digestive, hepatic and endocrine-metabolic diseases. Post-Surgical and Transplant Intensive Care Unit.

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
