# Peer review of "Planned Extracorporeal Life Support Employment during Liver Transplantation: The Potential of ECMO and CRRT as Preventive Therapies—Case Reports and Literature Review"

_jcm, 2023, doi:10.3390/jcm12031239_

Round 1
Reviewer 1 Report
A good attempt by the authors. However, there are some comments which need to be incorporated.
- If authors find it suitable they may add Case reports and a review of literature in the title of the paper. I think this addition justifies the content of the article.
- Under the section Material and Methods, the time period should be written as January 2021- January 2022 (Example) to understand the exact time period of the study.
- Though Diagrammatic representations about the selection of relevant literature are available, authors can write a brief about the inclusion and exclusion criteria of selections for both the cases and literature review under the section Material and Methods
- The clinical parameters value of all the cases can be written in tabular form under the respective section
- I have a suggestion for the author to add additional cost implications if this would be commonly used across the globe. I mean what was the additional cost burden on patients in such scenarios?
Reviewer 2 Report
Dear Authors,
Dear Editors,
The manuscript describes the sound management of three LT cases with relative LT contraindications or stuck in apparent vicious cycles, in which a judicious personalized, resource-intensive approach provided the ultimate curative solution. While there is little to argue from a medical management standpoint, the manuscript contains some inconsistencies.
Major:
- In section 3.4, including table 3 and Figure 2, the first case is referenced as a case of severe HPS, while the actual case report describes a case of moderate PoPH (with an initial mPAP < 50 mmHg), with no evidence of HPS (A-a gradient, shunting). Case 2 is referred to as PoPH + AKI, while the report describes a typical case of severe HPS (with bubble echo depiction of shunting). Finally, the third case is referred to as PoPH. While portal hypertension might be a contributor to PAH, in this case, I firmly believe that the main component is left-sided heart failure. Consequently, this is most likely a case of group 2 PAH occurring in a cirrhotic patient rather than pure type 1 PoPH. Either way, this bears little relevance for the actual case management, but given that the pulmonary complications of portal hypertension typically represent a gray area o expertise in most non-LT centers (and in some LT centers as well), I care quite a lot for the correct use of terms and classifications.
- I recommend a slightly more in-depth description of the conditions of these three patients. This should increase the educational value of the manuscript and clarify the indications for the therapeutic choice. Some of the following might help:
- Describing the full RHC report (mPAP, PAWP, PVR) and summarizing the report with a conclusion (i.e., "the RHC revealed an mPAP of X, with PAWP of Y and PVR of Z, highly suggestive for mild/moderate/severe type 1 PAH. Given the presence of portal hypertension and the lack of alternate causes for PAH - lung disease, left heart failure, the most probable diagnosis is moderate PoPH"). Neither of the three cases has a full RHC report described.
- Justifying the diagnostic approach (i.e., "given the peristent hypoxemia and an A-a gradient of... mmHg in a patient with cirrhosis and late decompensation - arguably, case 2 looks like ACLF - a CE-TTE or bubble echo was performed to confirm/rule out HPS. The CE-TTE revealed the presence of the bubbles in the LV within N cardiac cycles, thus confirming HPS")
- Justifying therapy - i.e. "while mild or moderate PoPH should generate MELD exception points and facilitate accession to LT, severe or unresponsive PoPH is a (relative) contraindication for LT. Thus, anti-PAH therapy was supplemented by increasing the dose of sildenafil and adding another agent. This led to a decrease in mPAP from a threshold value of 35, corresponding to moderate PoPH, to 32, corresponding to mild PoPH."
- Clearly defining the rationale for VV-ECMO or VA-ECMO.
- Describing the indication-specific outcomes (resolution of HPS, the persistence of PoPH, subsequent RHC, if available).
- Describe alternative approaches and expected outcomes (i.e. without using VV-ECMO in case 1, outcome X or Y would have been possible, VA-ECMO likely prevented outcome Z...)
Minor:
- I recommend referencing the most relevant literature on PoPH or HPS in LT and positioning these cases within the framework of the available guidelines (EASL, ILTS guidelines on PoPH, and HPS in LT).
- Figure 1 appears to be a PrintScreen from a PPT slide (one word is underlined, and the third box is highlighted). In figure 2, the ECMO boxes should state "if elevated" instead of "if elevate".
- Minor spelling and typo correction is required.
Reviewer 3 Report
Please see the insertions in the variant of attached manuscript; there are minor corrections (text editing) to consider.

Round 2
Reviewer 2 Report
I want to congratulate the authors on their work. I have no further comments.
Author Response
We thank Reviewer 2 for his opinion.